# Validation of Risk-Adapted Venous Thromboembolism Prediction in Multiple Myeloma Patients

**DOI:** 10.3390/jcm10163536

**Published:** 2021-08-12

**Authors:** Aisling Barrett, John Quinn, Michelle Lavin, Patrick Thornton, James O’Donnell, Philip Murphy, Siobhán Glavey

**Affiliations:** 1Department of Haematology, Beaumont Hospital, D09 V2N0 Dublin, Ireland; aibarret@tcd.ie (A.B.); johnquinn@beaumont.ie (J.Q.); patrickthornton@beaumont.ie (P.T.); philipmurphy@beaumont.ie (P.M.); 2Irish Centre for Vascular Biology, Royal College of Surgeons in Ireland, D02 VN51 Dublin, Ireland; michellelavin@rcsi.com (M.L.); jamesodonnell@rcsi.com (J.O.); 3Departments of Pathology and Haematology, Royal College of Surgeons in Ireland, D02 VN51 Dublin, Ireland

**Keywords:** multiple myeloma, venous thromboembolism, direct oral anticoagulants, thromboprophylaxis

## Abstract

Multiple myeloma (MM) is associated with an increased risk of venous thrombosis (VTE). In the United Kingdom Medical Research Council (MRC) XI study of patients treated with immunomodulatory therapy, the VTE rate was 11.8% despite 87.7% of the patients being on thromboprophylaxis at the time of thrombosis. In order to effectively prevent VTE events in MM patients, a better understanding of patient and disease risk factors that might predict thrombosis is required. We performed a retrospective cohort analysis of over 300 newly diagnosed MM patients at a tertiary referral centre to determine the VTE rate, predictive factors for VTE, value of the Khorana score for MM VTE events and long-term mortality outcomes. Fifty-four percent of the patients were receiving thromboprophylaxis at the time of the VTE event. The mortality odds ratio was 3.3 (95% CI, 2.4–4.5) in patients who developed VTE in comparison to age-matched controls with MM. A younger age at diagnosis and higher white cell count (WCC) were found to be predictive of VTE events. Our data suggest that standard thromboprophylaxis may not be effective in preventing VTE events in myeloma patients, and alternative strategies, which could include higher-intensity thromboprophylaxis in young patients with a high WCC, are necessary.

## 1. Introduction

Multiple myeloma (MM) is a plasma cell malignancy that confers up to a 9-fold increased risk of venous thromboembolism (VTE) [1]. The proposed risk factors for VTE development include patient-related factors such as a personal history of VTE, obesity, glucocorticoid use and comorbidities, and myeloma-related factors such as renal disease, hyperviscosity, immobilisation due to bone disease and treatment-related effects. The patients at highest risk include those treated with immunomodulatory drugs (IMiDs) such as thalidomide or lenalidomide, for whom the VTE rate has been reported to be as high as 11.8% in the United Kingdom Medical Research Council (MRC)-XI trial [2]. The IMiD-related VTE risk is also influenced by combination with high-dose glucocorticoids, with several studies demonstrating the incidence of VTE to be almost three times higher in lenalidomide and dexamethasone (RD)-treated patients than in those receiving dexamethasone alone [3,4].

Strategies for VTE prevention in MM patients have been adopted from the cancer literature and associated guidelines and are based on risk stratification with the use of low-molecular-weight heparin (LMWH) or aspirin according to risk factors. Evidence is accumulating, however, that these protocols may not be effective in identifying at-risk MM patients or in preventing VTE events [5]. This provides evidence that MM patients are a distinct population of cancer patients with likely alternative mechanisms of VTE development and risk factors at play. VTE has also been associated with inferior survival outcomes in myeloma patients, even in the novel therapy era, making a focus on predictive markers and the pathophysiologic mechanisms of thrombosis in these patients imperative [6].

The Khorana score has been shown to accurately predict cancer-associated thrombosis in non-haematological malignancies; however, it has been shown that the score is not predictive of VTE in MM patients [7,8]. In an effort to gain further knowledge on risk factors and VTE prevention strategies for these patients, we aimed to evaluate the utility of this score in MM patients.

## 2. Materials and Methods

We performed a retrospective cohort analysis of all newly diagnosed MM patients at our institution over a 16-year period to determine the predictive value of the Khorana score for MM VTE events. The primary objective was to examine the predictive value of age, white cell count (WCC) and haemoglobin (Hb) at the diagnosis of MM for subsequent VTE events. Secondary predictive factors were also examined, which included myeloma-specific markers (beta-2-microglobulin and paraprotein burden and subtype), MM therapy at the time of VTE, other risk factors for VTE and thromboprophylaxis at the time of VTE. Patients aged 18 and above presenting with a new diagnosis of MM between 2001 and 2017 were included in the study. Medical notes, radiology reports and laboratory tests were reviewed, and patients with radiologically confirmed deep vein thrombosis (DVT) or pulmonary embolus (PE) were included in the analysis. Patients presenting with primary plasma cell leukaemia or who presented acutely with VTE at the time of initial MM presentation were excluded. Statistical analysis was performed using logistic regression and Cox proportional hazard modelling.

## 3. Results

A total of 332 MM patients met the inclusion criteria and had their data analysed. Thirty-two patients (9.6%) were diagnosed with VTE, with a median time from myeloma diagnosis to VTE occurrence of 13.5 months (11 days to 11.8 years). PE occurred in 14 (44% of VTEs) patients, with two being bilateral and extensive. Eighteen patients (56% of VTEs) were diagnosed with DVT, of which seven were considered extensive in the iliofemoral circulation. Table 1 demonstrates the therapy regimens of 24 patients (92.3%) with these data available at the time of VTE diagnosis. The median number of lines of treatment prior to VTE development was 2 (0–5). Nine patients (39% of the 23 for whom these data were available) suffered VTE during their induction chemotherapy. Twelve (38%) patients had other risk factors for VTE at the time of thrombosis, the majority of which were transient or acute, and five (15%) patients had more than one VTE event during the 16-year follow-up period.

The mortality odds ratio was 3.3 (95% CI, 2.4–4.5) in patients who developed VTE in comparison to age-matched controls with MM, demonstrating this as an independent risk factor for early death in MM. Importantly, no fatal cases of VTE occurred in our cohort, indicating that VTE specifically confers inferior MM outcomes independent of traditional MM markers. The majority of mortality in these cases was linked to disease progression and infection with active disease. A younger age at diagnosis of MM was found to be predictive of VTE development in univariate (*p* = 0.002) and multivariate (*p* = 0.004) analysis (a mean age of 63.8 years in the VTE cohort versus 68.6 years in the non-VTE cohort), and this finding was independent of the MM duration. When individual variables from the Khorana score were subjected to univariate and multivariate analysis in our cohort, the WCC was the only variable that retained predictive significance, with a higher WCC at MM diagnosis showing a trend towards significance in univariate analysis (*p* = 0.06) for predicting VTE. Neither the paraprotein subtype or burden, beta-2-microglobulin level or International Myeloma Working Group (IMWG) international staging system (ISS) status at diagnosis were predictive of VTE. The baseline characteristics of the VTE and non-VTE populations are outlined in Table 2.

A total of 54% of the patients for whom data were available were receiving thromboprophylaxis at the time of VTE. Table 3 outlines the thromboprophylaxis regimens received by patients at the time of VTE. Twelve patients (32.4%) had other risk factors for VTE at the time of thrombosis.

Eleven patients were not receiving thromboprophylaxis at the time of the development of VTE. Six patients of the 11 (54.5%) were on regimens that did not include an immunomodulatory agent, were clinically risk-assessed for VTE prophylaxis and were deemed not at high risk of VTE; one patient was thrombocytopaenic due to autologous stem cell transplantation; one was undergoing stem cell harvest; one had just stopped treatment with thalidomide, and his thromboprophylaxis had accordingly been withdrawn; in two cases, thromboprophylaxis was omitted with no clear explanation for the same.

## 4. Conclusions

Our data confirm that the development of VTE is associated with reduced overall survival in MM independent of the disease course or stage at presentation. An age of less than 64 at diagnosis in combination with a high WCC may be the most likely simple predictive combination for VTE in MM, with none of the other components of the Khorana score reaching significance in our cohort. The association between a younger age and higher risk of VTE has been observed in other studies [7]. In our study, proportionally more patients in the over-65 age group were receiving IMiDs with a known risk of VTE. We feel the higher rate of VTE in younger patients in our study is therefore not likely to have been related to the therapeutic regimens, and in these patients other physiological factors may have been important in VTE risk. This requires further investigation in randomised settings.

The findings of a sub-group analysis of the MRC-XI trial [2], which showed that the rate of VTE was up to 11.8% in a cohort of MM patients with 80.5% receiving thromboprophylaxis, are supported by our study. However, the rate of thromboprophylaxis specifically amongst those who subsequently developed VTE was not specified, and aspirin was used in one-third of patients receiving thromboprophylaxis. Guidelines suggest that those undergoing therapy with an IMiD with one risk factor or less should receive aspirin as primary thromboprophylaxis, with other patients recommended to receive LMWH [5]. The IMPEDE-VTE score aims to improve on the current risk stratification for VTE in MM and includes therapy with an IMiD, body mass index, pathologic fractures, treatment with erythropoiesis-stimulating agents, dexamethasone or doxorubicin therapy, ethnicity, history of VTE, the presence of an indwelling tunnelled line and existing thromboprophylaxis [9]. The 6-month cumulative incidence of VTE for scores of ≥8 was 15.2; however, this study did not include patients under the age of 65 in the validation cohort and therefore may not reflect specific VTE risks in this age group. This score has also been validated in the MATISSE database, which included MM patients older than 18 years [10].

The mechanism of VTE development in MM is likely multifactorial and influenced by the disease, therapy and patient-specific factors, and a greater understanding of the pathophysiology is required to inform alternative mechanisms of thrombosis prevention in MM patients. Pro-thrombotic abnormalities that predispose MM patients to VTE such as elevated factor VIII levels, acquired activated protein C resistance, and hypofibrinolysis are potential areas for future investigation [11]. Tissue factor (TF) expression has also been shown to contribute to VTE risk in malignancy [12], and interestingly, persistently elevated levels of TF have been observed in MM patents following chemotherapy who subsequently developed a thrombosis [13].

Alternate thromboprophylaxis strategies for MM under consideration at present include the use of direct oral anticoagulants (DOACs). Data are accumulating regarding the use of apixaban in primary VTE prevention in MM patients treated with immunomodulatory agents. Three recent studies comprising 224 patients in total have evaluated VTE and bleeding rates with the use of apixaban at 2.5 mg twice daily for at least 6 months, with no recorded VTE events while on anticoagulation (two events were recorded after the cessation of anticoagulation due to medication-induced thrombocytopaenia). This apparent efficacy may come with an increased bleeding risk, as the pooled data reveal two episodes of major haemorrhage and 14 episodes of clinically relevant non-major haemorrhage [14,15,16]. Although the data are not validated in comparison to another method of thromboprophylaxis in a randomised trial, the results of this study showing no VTE events in any patient actively using apixaban are encouraging.

The limitations of our study include the small sample size and the retrospective nature of the data collection, and the findings will require validation in larger cohorts. Data regarding VTE prophylaxis were only available for 24 of 32 patients (75%) who developed VTE. No data on the use of erythropoiesis-stimulating agents were collected. The small sample size for patients on LMWH is a limitation of our study, and therefore we cannot draw conclusions on the efficacy of this regimen in myeloma patients. No patient diagnosed with thrombosis in our study had been treated with daratumumab, an anti-CD138 therapy that is now licensed for induction therapy for MM as well as for the relapsed setting by the European Medicines Agency (EMA). This is likely reflective of poor access to the medication prior to the data collection cut-off. A recent post hoc analysis of three large trials revealed no statistically significant difference in VTE rates in patients receiving daratumumab compared to those receiving other therapies [17].

We conclude that patients with MM should be fully risk assessed for VTE at diagnosis and that this is dynamically reassessed throughout their disease course, with possible consideration of higher-intensity thromboprophylaxis in young patients with a high WCC. 

## Figures and Tables

**Table 1 jcm-10-03536-t001:** Chemotherapeutic regimens of patients at time of VTE diagnosis. VAD = vincristine/doxorubicin/dexamethasone.

Chemotherapy at Time of Thrombosis	Number of Patients (*n* = 24)
Lenalidomide	10 (41.7%)
Thalidomide	4 (16.7%)
Melphalan	3 (12.5%)
At time of stem cell harvest or reinfusion	2 (8.3%)
Bortezomib	1 (4.2%)
Thalidomide/cyclophosphamide	1 (4.2%)
Pre-treatment	1 (4.2%)
Older regimens (VAD, idarubicin)	2 (8.3%)
Plus high-dose steroid therapy	17 (70.8%)

**Table 2 jcm-10-03536-t002:** Baseline characteristics of the VTE and non-VTE populations. Hb = haemoglobin. WCC = white cell count.

	VTE Population (*n* = 32)	Non-VTE Population (*n* = 300)
Male	22 (68.8%)	188 (62.7%)
Female	10 (31.3%)	112 (37.3%)
Median age at MM diagnosis (years)	63.9 years (39.9–83.9)	70.7 years (34.7–89.0)
Number of patients <65 years old	20 (62.5%)	85 (28.3%)
Median Hb concentration at diagnosis (g/dL)	9.65 (7.3–15)	10.1 (5.9–15.6)
Median WCC at diagnosis (×10^9^/L)	7.4 (1.95–21.4)	6.4 (1.8–45.4)
Median platelet count at diagnosis (×10^9^/L)	218.5 (114–772)	227.5 (7–782)
Beta-2 microglobulin level at diagnosis (µg/L)	4.4 (1.7–33.5)	5.1 (1.4–76.2)
Paraprotein level at diagnosis (g/L)	33.9 (<1–77.1)	25.5 (0–104)

**Table 3 jcm-10-03536-t003:** Prophylaxis regimens at time of development of VTE. LMWH = low-molecular-weight heparin. VKA = vitamin K antagonist.

Thromboprophylaxis	Number of Patients (*n* = 24)
No prophylaxis	11 (45.8%)
Aspirin	6 (25%)
Low-molecular-weight heparin	6 (25%)
Vitamin K antagonist	1 (4.2%)

## Data Availability

The data presented in this study are available on request from the corresponding author.

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
