# Peer review of "Validation of Risk-Adapted Venous Thromboembolism Prediction in Multiple Myeloma Patients"

_jcm, 2021, doi:10.3390/jcm10163536_

Round 1
Reviewer 1 Report
Abstract : the VTE rate was 11.8% despite 87.7% of patients being on thromboprophylaxis at the time of thrombosis : 12% without thromboprophylaxis…
Could we considered aspirin as a real thromboprophylaxis prevention ??? I dont think so and so 87.7% of aptients are not under thromboprophylaxis
« 54% of patients for whom data was available » (laking data ?) « were receiving thromboprophylaxis at the time of VTE. » 25% under aspirin…
« Of the 46% not receiving prophylaxis reasons included that the patient did not meet risk criteria for same, were thrombocytopenic or had an increased risk of bleeding » which line of treatment ? not half of patients with multiple myeloma were thrombocytopenic and with increased risk (specify reasons !)
« the hypothesis that more intensive chemotherapeutic regimens could contribute to increased VTE risk in a fitter younger patient is not borne out by our study where the majority of these patients received single/double-agent plus dexamethasone regimens and proportionally more (69.2% versus 50%) patients in the over 65-years age cohort were being treated with IMiDs at the time of thrombosis. » so this is a confunding bias and not because patients are young !!! wrong message
What about EPO ?
Line of treatment ?
Time of VTE from treatment starting ?
IMDE VTE should be use « this study did not include patients under the age of 65 in the validation cohort » yes in the « external » validation cohort but there are un validation cohort. This was confirmed in a letter from Am J Hem with an other cohort including young people (melisse database).
Khorana score ? and what about the specific MM khorana score ?
judge not effective « Our data suggest that LMWH thromboprophylaxis may not be effective in preventing VTE events in myeloma patients and alternative strategies are necessary » ? only 6 patients with vte…
apixaban ? not in first line, not comparative etc :
- Storrar NPF, Mathur A, Johnson PRE et al. Safety and efficacy of apixaban for routine thromboprophylaxis in myeloma patients treated with thalidomide- and lenalidomide-containing regimens. Br J Haematol 2019;185(1):142-4. doi: 10.1111/bjh.15392. 13. Cornell RF, Goldhaber SZ, Engelhardt BG et al. Primary prevention of venous thromboembolism with apixaban for multiple myeloma patients receiving immunomodulatory agents. Br J Haematol 2020 Aug;190(4):555-561. doi: 10.1111/bjh.16653. 14. Pegourie B, Karlin L, Benboubker L et al. Apixaban for the prevention of thromboembolism in immunomodulatory- treated myeloma patients: Myelaxat, a phase 2 pilot study. Am J Hematol 2019 Jun;94(6):635-640. doi: 10.1002/ajh.25459
Author Response
July 11th, 2021
Re: Barrett et al: Younger Age at Diagnosis is Associated with an Increased Risk of Venous Thromboembolism in Multiple Myeloma
To:
Dr E Andrès
Editor-in-Chief of Journal of Clinical Medicine
Prof S Rutella
Hematology Section Editor-in-Chief of Journal of Clinical Medicine
Many thanks for considering the publication of this manuscript following major revisions in the Journal of Clinical Medicine.
In this manuscript we present an analysis of over 300 multiple myeloma (MM) patients at a tertiary referral center and describe risk factors for venous thromboembolism (VTE) development over a 16-year period. We demonstrate that cancer VTE predictive scores such as the Khorana score are not predictive of VTE development in these patients and that thromboprophylaxis is often not effective in reducing VTE events. We also perform multivariate analysis demonstrating that younger age and higher white cell count (WCC) at diagnosis are associated with an increased risk of VTE in these patients and that this may serve as a simple scoring system to identify patients at higher risk.
We thank the reviewers for their constructive comments and have made major revisions to the manuscript to reflect their suggested edits. In the below rebuttal we address each comment individually and the manuscript has been marked up to clearly outline where changes have been made. The suggested changes and edits have increased the quality of our manuscript and we are hopeful that it will now be acceptable for publication in the Journal of Clinical Medicine.
With kindest regards,
Professor Siobhan Glavey
Corresponding & Senior Author of Manuscript
Departments of Pathology and Haematology
Royal College of Surgeons in Ireland and Beaumont Hospital, Dublin, Ireland
REVIEWER ONE
- Abstract : the VTE rate was 11.8% despite 87.7% of patients being on thromboprophylaxis at the time of thrombosis : 12% without thromboprophylaxis…
This point is well made by the reviewer. This analysis of patients with thrombosis on the MRC-XI study does not specify whether the patients who developed thrombosis were exclusively the patients who were not receiving thromboprophylaxis at the time of their event. We have amended the manuscript to include the fact that 12% of patients in this study were not on VTE prophylaxis – line 160-163.
- Could we considered aspirin as a real thromboprophylaxis prevention ??? I dont think so and so 87.7% of patients are not under thromboprophylaxis
The reviewers correctly point out that aspirin is not a standard of care for effective VTE prophylaxis. As per the Bradbury et al. article, 33.7% of the patients in the MRC-XI study were on aspirin as thromboprophylaxis, 4% were on treatment-dose warfarin and 2% were on “another” anticoagulant but the majority (60.3%) were on low-molecular-weight heparin (LMWH) as thromboprophylaxis– we have included this information in the manuscript line 163-164.
- « 54% of patients for whom data was available » (lacking data ?) « were receiving thromboprophylaxis at the time of VTE. » 25% under aspirin…
The reviewer correctly points out that data regarding VTE prophylaxis was available in only 24 of 32 patients (75%) for analysis. We recognise this as a limitation in our study on lines 199-200. However, given that this was a retrospective data collection of patient data over 16 years we feel this is important to acknowledge as it reflects real world data outside of a trial setting.
- « Of the 46% not receiving prophylaxis reasons included that the patient did not meet risk criteria for same, were thrombocytopenic or had an increased risk of bleeding » which line of treatment ? not half of patients with multiple myeloma were thrombocytopenic and with increased risk (specify reasons!)
6 patients of the 11 (54.5%) were on regimens that did not include an immunomodulatory agent and were clinically risk-assessed for VTE prophylaxis and were deemed not at high risk of VTE. Of the other five: 1 was thrombocytopaenic due to autologous stem cell transplantation, 1 was peri-stem cell harvest, 1 had just stopped treatment with thalidomide and his thromboprophylaxis had accordingly been withdrawn, and in two cases thromboprophylaxis was omitted with no clear explanation for same. These reasons have been outlined in lines 134-141.
- « The hypothesis that more intensive chemotherapeutic regimens could contribute to increased VTE risk in a fitter younger patient is not borne out by our study where the majority of these patients received single/double-agent plus dexamethasone regimens and proportionally more (69.2% versus 50%) patients in the over 65-years age cohort were being treated with IMiDs at the time of thrombosis. » so this is a confunding bias and not because patients are young !!! wrong message
We agree with this statement by the reviewer and have amended the article accordingly on lines 154-158 to state “Proportionally more patients in the over 65 age group were receiving IMIDs with a known risk of VTE. In our study the higher rate of VTE in younger patients is not likely to have been related to therapeutic regimens, and in these patients other physiological factors may have been important in VTE risk. This requires further investigation in randomised settings”.
- What about EPO?
There were no data collecting regarding the use of erythropoietin, which is a limitation of the study – we acknowledge this limitation in lines 200-201. We do not have a high usage rate for EPO in our clinical practice and therefore we do not believe this was a factor in VTE risk, however our study did not assess this.
- Line of treatment?
The median number of lines of treatment prior to VTE development was 2 (0-5). 9 patients (39% of the 23 for whom these data were available) suffered VTE during their induction chemotherapy – this has now been included in the results section in lines 93-95.
- Time of VTE from treatment starting?
The median time from MM diagnosis to VTE development is outlined in the manuscript but data regarding the time of VTE from commencement of the specific line of therapy were not consistently collected.
- IMPEDE VTE should be use « this study did not include patients under the age of 65 in the validation cohort » yes in the « external » validation cohort but there are un validation cohort. This was confirmed in a letter from Am J Hem with another cohort including young people (melisse database).
We thank the reviewer for drawing our attention to these data which show similar results of validation of the score in a population of MM patients older than 18 years old.1 This is now included in the manuscript in lines 173-174 along with the below reference.
- Chalayer E et al. Am J Hematol. 2020 Jan;95(1):E18-E20.
- Khorana score? And what about the specific MM Khorana score?
When individual variables from this score were subjected to univariate and multivariate analysis in our cohort, white cell count was the only variable that retained predictive significance. We have amended the results to more clearly reflect this on lines 111-113.
- Judge not effective « Our data suggest that LMWH thromboprophylaxis may not be effective in preventing VTE events in myeloma patients and alternative strategies are necessary » ? only 6 patients with vte…
We agree with the reviewer’s comment and have amended the manuscript on lines 201-203.
- Apixaban ? not in first line, not comparative etc.
We agree with the reviewer’s comment and have amended the manuscript accordingly in lines 195-197- although the data are not comparative to another method of thromboprophylaxis, the results of this study, showing no VTE events in any patient actively using apixaban are encouraging.

Reviewer 2 Report
This is a well written short article by the group of Prof Siobhan Glavey on an important topic in the management of patients with Multiple Myeloma (MM). In MM, several therapies have been developed in the last years with improving in disease-free survival and overall survival for these patients. Among the commonest complications seen in this population is venous thromboembolism (VTE), as more than 10% will develop VTE during the course of their disease.
The Khorana score (Blood 2008) has been shown to accurately predict cancer-associated thrombosis in cancers and the authors. In this respect the authors, in the current paper, studied risk factors and VTE prevention strategies for MM patients, evaluating also the utility of Khorana score in MM patients.
The paper is well written, but there are some comments to make, suggesting important implementations or clarifications
Major comments:
- The authors stated that Khorana score has been shown to accurately predict cancer-associated thrombosis in non-hematological malignancies, however it has not been validated in MM patients; it is false since in the original paper (Blood 2008), 12.1% and 13.5% in derivation and validation cohorts, respectively, had a diagnosis of Lymphoma. Moreover, Khorana score was previously assessed in a large cohort of 2874 patients with MM (Abstract at ASCO 2018 entitled “Predictive ability of the Khorana score for venous thromboembolism (VTE) in multiple myeloma (MM)”, Sanfilippo et al.); the authors concluded that the Khorana score did not accurately predict VTE in a MM population. This is the aim of the current study and should be clarified both in introduction and conclusions.
- A table with patients’ characteristics (n=322) and differences for patients with and without VTE must be include.
- Univariate and multivariable analyses for VTE must be also outlined in a table.
Author Response
July 11th, 2021
Re: Barrett et al: Younger Age at Diagnosis is Associated with an Increased Risk of Venous Thromboembolism in Multiple Myeloma
To:
Dr E Andrès
Editor-in-Chief of Journal of Clinical Medicine
Prof S Rutella
Hematology Section Editor-in-Chief of Journal of Clinical Medicine
Many thanks for considering the publication of this manuscript following major revisions in the Journal of Clinical Medicine.
In this manuscript we present an analysis of over 300 multiple myeloma (MM) patients at a tertiary referral center and describe risk factors for venous thromboembolism (VTE) development over a 16-year period. We demonstrate that cancer VTE predictive scores such as the Khorana score are not predictive of VTE development in these patients and that thromboprophylaxis is often not effective in reducing VTE events. We also perform multivariate analysis demonstrating that younger age and higher white cell count (WCC) at diagnosis are associated with an increased risk of VTE in these patients and that this may serve as a simple scoring system to identify patients at higher risk.
We thank the reviewers for their constructive comments and have made major revisions to the manuscript to reflect their suggested edits. In the below rebuttal we address each comment individually and the manuscript has been marked up to clearly outline where changes have been made. The suggested changes and edits have increased the quality of our manuscript and we are hopeful that it will now be acceptable for publication in the Journal of Clinical Medicine.
With kindest regards,
Professor Siobhan Glavey
Corresponding & Senior Author of Manuscript
Departments of Pathology and Haematology
Royal College of Surgeons in Ireland and Beaumont Hospital, Dublin, Ireland
REVIEWER TWO
- The authors stated that Khorana score has been shown to accurately predict cancer-associated thrombosis in non-hematological malignancies, however it has not been validated in MM patients; it is false since in the original paper (Blood 2008), 12.1% and 13.5% in derivation and validation cohorts, respectively, had a diagnosis of Lymphoma.
The reviewer correctly points out that some patients with haematological malignancy were included in the original validation studies of the Khorana score– however no patients with multiple myeloma were included in the published data. Given the differences in clinical risk of VTE with myeloma treatment regimens, we feel this group of patients deserve separate attention.
- Moreover, Khorana score was previously assessed in a large cohort of 2874 patients with MM (Abstract at ASCO 2018 entitled “Predictive ability of the Khorana score for venous thromboembolism (VTE) in multiple myeloma (MM)”, Sanfilippo et al.); the authors concluded that the Khorana score did not accurately predict VTE in a MM population. This is the aim of the current study and should be clarified both in introduction and conclusions.
We are grateful to the reviewer for drawing our attention to this important study which showed that the Khorana score was not predictive of thrombosis in a specific MM cohort.2 We have now include this study in the manuscript on lines 66-68 with the below reference.
- Sanfilippo KM et al. Journal of Clinical Oncology. 36, no. 15_suppl.
- A table with patients’ characteristics (n=322) and differences for patients with and without VTE must be included.
We thank the reviewer for this useful suggestion which we feels adds to the interpretation of our study. See new table below.
|
|
VTE Population (n=32) |
Non-VTE Population (n=300) |
|
Male |
22 (68.8%) |
188 (62.7%) |
|
Female |
10 (31.3%) |
112 (37.3%) |
|
Median age at MM diagnosis (years) |
63.9 years (39.9-83.9) |
70.7 years (34.7-89.0) |
|
Number of patients <65 years old |
20 (62.5%) |
85 (28.3%) |
|
Median Hb concentration at diagnosis (g/dL) |
9.65 (7.3-15) |
10.1 (5.9-15.6) |
|
Median WCC at diagnosis (x109/L) |
7.4 (1.95-21.4) |
6.4 (1.8-45.4) |
|
Median platelet count at diagnosis (x109/L) |
218.5 (114-772) |
227.5 (7-782) |
|
Beta-2 microglobulin level at diagnosis (ųg/L) |
4.4 (1.7-33.5) |
5.1 (1.4-76.2) |
|
Paraprotein level at diagnosis (g/L) |
33.9 (<1-77.1) |
25.5 (0-104) |
- Univariate and multivariable analyses for VTE must be also outlined in a table.
We thank the reviewer for this comment, these data are clearly outlined in the manuscript– inclusion of this table will likely exceed the journal limits but we are happy to do so if the editor welcomes this edit to the manuscript.

Reviewer 3 Report
The work is interesting, although I have many concerns.
1) It's necessary to specify the age considered as cut off to define a patient with multiple myeloma "younger"
2) While the therapies of patients with the thromboembolic events are described, in my opinion is important to describe the treatment of the whole cohort
3) Moreover, is important to me to better define mortality causes of patients with VTE.
4) The number of patients is low for definitive conclusions. Probably the authors should modify the title, as the aim of the study is interesting.
Author Response
July 11th, 2021
Re: Barrett et al: Younger Age at Diagnosis is Associated with an Increased Risk of Venous Thromboembolism in Multiple Myeloma
To:
Dr E Andrès
Editor-in-Chief of Journal of Clinical Medicine
Prof S Rutella
Hematology Section Editor-in-Chief of Journal of Clinical Medicine
Many thanks for considering the publication of this manuscript following major revisions in the Journal of Clinical Medicine.
In this manuscript we present an analysis of over 300 multiple myeloma (MM) patients at a tertiary referral center and describe risk factors for venous thromboembolism (VTE) development over a 16-year period. We demonstrate that cancer VTE predictive scores such as the Khorana score are not predictive of VTE development in these patients and that thromboprophylaxis is often not effective in reducing VTE events. We also perform multivariate analysis demonstrating that younger age and higher white cell count (WCC) at diagnosis are associated with an increased risk of VTE in these patients and that this may serve as a simple scoring system to identify patients at higher risk.
We thank the reviewers for their constructive comments and have made major revisions to the manuscript to reflect their suggested edits. In the below rebuttal we address each comment individually and the manuscript has been marked up to clearly outline where changes have been made. The suggested changes and edits have increased the quality of our manuscript and we are hopeful that it will now be acceptable for publication in the Journal of Clinical Medicine.
With kindest regards,
Professor Siobhan Glavey
Corresponding & Senior Author of Manuscript
Departments of Pathology and Haematology
Royal College of Surgeons in Ireland and Beaumont Hospital, Dublin, Ireland
1) It's necessary to specify the age considered as cut off to define a patient with multiple myeloma "younger".
Our data demonstrated a mean age of 63.8 years in VTE cohort vs. 68.6 years in non-VTE cohort, therefore this may indicate that in patients under the age of 64 years are at increased risk of VTE. We have more clearly outlined this in the manuscript on lines 151.
2) While the therapies of patients with the thromboembolic events are described, in my opinion is important to describe the treatment of the whole cohort
As the study focused specifically on patients with VTE, we did not collect data on the non-VTE cohort. We do not believe that the treatment data in this non-VTE cohort should be different overall to the VTE cohort, given that the treatments available to treating clinicians within the time frame of the study, would have been limited to approved regimens. We agree with the reviewer that it would be interesting to investigate if there were any differences in treatments used, but we feel this is unlikely to be the case.
3) Moreover, is important to me to better define mortality causes of patients with VTE.
For the patients with VTE the majority of mortality in these cases was related to disease progression and infection with active disease. This information has now been added to the manuscript on line 110-111.
4) The number of patients is low for definitive conclusions. Probably the authors should modify the title, as the aim of the study is interesting.
We agree with the reviewer’s comments and have altered the title on lines 3-4 of the manuscript as follows: “Validation of risk-adapted venous thromboembolism prediction in multiple myeloma patients”.
Round 2
Reviewer 1 Report
Despite revisions, this article seems contain a low scientific interest
Author Response
August 4th, 2021
Re: Barrett et al: Younger Age at Diagnosis is Associated with an Increased Risk of Venous Thromboembolism in Multiple Myeloma
To:
Dr E Andrès
Editor-in-Chief of Journal of Clinical Medicine
Prof S Rutella
Hematology Section Editor-in-Chief of Journal of Clinical Medicine
Many thanks for considering the publication of this manuscript following major revisions in the Journal of Clinical Medicine.
In this manuscript we present an analysis of over 300 multiple myeloma (MM) patients at a tertiary referral center and describe risk factors for venous thromboembolism (VTE) development over a 16-year period. We thank the reviewer for their comments. We believe our data to be useful in demonstrating that cancer VTE predictive scores such as the Khorana score are not predictive of VTE development in these patients and that thromboprophylaxis is often not effective in reducing VTE events. We also perform multivariate analysis demonstrating that younger age and higher white cell count (WCC) at diagnosis are associated with an increased risk of VTE in these patients and that this may serve as a simple scoring system to identify patients at higher risk. We acknowledge the limitations of our study including the small sample size and the retrospective nature of data collection. We believe that our study will aid future research including validation of our findings in larger cohorts.
The suggested changes and edits have increased the quality of our manuscript and we are hopeful that it will now be acceptable for publication in the Journal of Clinical Medicine.
With kindest regards,
Professor Siobhan Glavey
Corresponding & Senior Author of Manuscript
Departments of Pathology and Haematology
Royal College of Surgeons in Ireland and Beaumont Hospital, Dublin, Ireland
Reviewer 2 Report
I read with interest the revised version of the paper "Younger Age at Diagnosis is Associated with an Increased Risk of Venous Thromboembolism in Multiple Myeloma". I state that the main points raised during the previous review have been amply expanded by the authors. I recommed to accept the manuscript in the current form.
Author Response
July 11th, 2021
Re: Barrett et al: Younger Age at Diagnosis is Associated with an Increased Risk of Venous Thromboembolism in Multiple Myeloma
To:
Dr E Andrès
Editor-in-Chief of Journal of Clinical Medicine
Prof S Rutella
Hematology Section Editor-in-Chief of Journal of Clinical Medicine
Many thanks for considering the publication of this manuscript following major revisions in the Journal of Clinical Medicine.
We thank the reviewers for their constructive comments and have made major revisions to the manuscript to reflect their previously suggested edits. The suggested changes and edits have increased the quality of our manuscript and we are hopeful that it will now be acceptable for publication in the Journal of Clinical Medicine.
With kindest regards,
Professor Siobhan Glavey
Corresponding & Senior Author of Manuscript
Departments of Pathology and Haematology
Royal College of Surgeons in Ireland and Beaumont Hospital, Dublin, Ireland
Reviewer 3 Report
The revisions made are fine